# The GHZ Theorem Revisited within the Framework of Gauge Theory

David H. Oaknin

Rafael Ltd., Haifa IL-31021, Israel; d1306av@gmail.com

**Abstract:** The Greenberger-Horne-Zeilinger version of the Einstein-Podolsky-Rosen (EPR) paradox is widely regarded as a conclusive logical argument that rules out the possibility of reproducing the predictions of Quantum Mechanics within the framework of any physical theory sharing the notions of reality and relativistic causality that we acknowledge as a given in our classical descriptions of the macroscopic world. Thus, this renowned argument stands as a seemingly insurmountable roadblock on the path to a very desired, physically intuitive understanding of quantum phenomena and, in particular, quantum entanglement. In this paper, we notice, however, that the GHZ argument involves unaccounted spurious gauge degrees of freedom and that it can be overcome once these degrees are properly taken into account. It is then possible to explicitly build a successful statistical model for the GHZ experiment based on the usual notions of relativistic causality and physical reality. This model, thus, completes—in the EPR sense—the quantum description of the GHZ state and paves the way to a novel intuitive interpretation of the quantum formalism and a deeper understanding of the physical reality that it describes.

**Keywords:** quantum mechanics; EPR paradox; Bell's theorem; GHZ argument; gauge symmetries; holonomies; hidden variables; statistical physics

## 1. Introduction

The inability to accommodate the seemingly trivial notions of causality and physical realism within the current interpretation of the quantum mechanical wavefunction is at the core of a long lasting debate about the foundations of quantum theory and the role played by measurements, whose origins go back to the formulation of the renowned Einstein-Podolsky-Rosen (EPR) paradox almost ninety years ago [1,2]. Solving these key issues would require developing a description of quantum phenomena in terms of a statistical model of local hidden variables. Nonetheless, according to the current wisdom, such a description is not possible in so far as we insist on keeping the notion, also seemingly trivial, that the observers' choice of their measurement settings is not constrained by the actual hidden configuration of the observed system (free-will).

Indeed, several fundamental theorems state that generic models of hidden variables that share certain intuitive features cannot fully reproduce the predictions of quantum mechanics [3–12], while carefully designed experimental tests have consistently confirmed the predictions of the quantum theory and, thus, have ruled out all these generic models of hidden variables [13–23]. The best known among these theorems is the Bell theorem [3,5–7], which proves that such generic models of hidden variables cannot reproduce the statistical correlations predicted by quantum mechanics for the outcomes of long sequences of strong polarization measurements performed along certain relative directions on pairs of entangled qubits.

The Greenberger-Horne-Zeilinger version of the Bell theorem [8] is an even more conclusive proof of the limitations of these generic models of hidden variables, since it proves that such models cannot reproduce even single outcomes of strong spin polarization

measurements performed along certain relative directions on three or more entangled qubits prepared in the so-called GHZ state,

$$|\text{GHZ}\rangle \equiv \frac{|\uparrow\uparrow \cdots \uparrow\rangle + |\downarrow\downarrow \cdots \downarrow\rangle}{\sqrt{2}}. \tag{1}$$

However, in a series of recent papers [24–27], we have shown that the proof of the Bell theorem crucially relies on a subtle assumption that is not required by fundamental physical principles. Namely, we noted that the proof implicitly assumes the existence of an absolute angular frame of reference with respect to which we can define the polarization properties of the hypothetical hidden configurations of the pairs of entangled qubits as well as the orientations of the measurement devices that test them. Furthermore, we showed that such an absolute frame of reference may not exist if the hidden configurations spontaneously break the gauge rotational symmetry along an otherwise arbitrary direction.

A simple example that illustrates the absence of an absolute frame is described in Figure 3 in reference [24]. Let us first consider a Bell-type game played between three parties located at the vertices of a triangle drawn on a plane. At the start of the game, each party sets at his/her vertex a reference unit vector contained within the plane. A long sequence of unit vectors randomly oriented within the plane is then produced at the center of each of the three edges of the triangle and sent to the two parties located at their respective ends. Upon receiving a sampled random vector, each party compares its orientation to the local reference unit vector and produces a binary outcome, either $+1$ or $-1$, according to a deterministic response function. In this game, the affine structure of the euclidean plane allows comparing at once the relative orientations of the reference unit vectors at the three vertices, as well as the sampled random unit vectors, and, thus, it defines an "absolute frame of reference". In precise terms, the plane is equipped with an equivalence relationship that allows it to univocally define the relative orientation of vectors located at different sites. It is then straightforward to derive the Bell inequality for the pairwise correlations between the binary outcomes of the parties. However, it can be readily seen that such an "absolute frame of reference" does not exist if we consider a similar Bell-type game played between parties located on the surface of a sphere instead of a plane: a tangent vector parallel-transported over a closed-loop drawn on the sphere may acquire a non-zero geometric rotation phase due to a holonomy. Therefore, even though any two parties can calibrate and agree on a common frame of reference to describe the relative orientations of their reference unit vectors as well as the orientation of the random vectors shared between them, there does not exist a common frame of reference upon which all three parties can agree at once. In order to compare (and maybe constrain) the pairwise correlations that can be attained in the latter game, it is necessary to set the reference unit vector of one of the parties as a fixed common frame by taking advantage of the gauge degrees of freedom involved in the problem.

Gauge degrees of freedom are auxiliary degrees that may appear in the theoretical models but do not correspond to well-defined degrees of freedom in the described physical system, so that the predictions of the model cannot depend on them [28]. In fact, theoretical models that involve spurious gauge degrees of freedom may require a gauge-fixing condition in order to make physically sound predictions. In a Bell experiment, the relative orientation between the two detectors that test the pairs of entangled qubits is a well-defined physical degree of freedom that actually determines the correlation between their outcomes. On the other hand, the global orientation associated with a rigid rotation of the two detectors is a spurious gauge degree of freedom that should not play any role in the predictions of any properly defined theoretical model. Similarly, the setting of the three detectors needed to test the triplets of qubits prepared in the GHZ state is described by a single physical degree of freedom too, as we shall show later.

Following these insights, we built in [24,25,27] an explicit statistical model of local hidden variables that fully reproduces the predictions of quantum mechanics for the Bell states of two entangled qubits while complying with all the required symmetry demands

and the hypothesis of 'free-will'. Thus, our model completes the description of these quantum states in the sense advocated by Einstein, Podolsky, and Rosen [1]. However, the model has been criticized because, even though it strictly complies with Einstein's causality principle, it supposedly violates Bell's definition of locality. In this respect, it is necessary to remember that Einstein's causality is a fundamental principle in modern physics that stems from the Lorentz covariance of the laws that describe the elementary building blocks of Nature and their interactions, while Bell's notion of locality arose only as a result of his intent to formulate Einstein's principle of causality in a way fit to prove his renowned theorem [3]. Therefore, wherever Einstein's causality principle and Bell's notion of locality do not agree, compliance with the former must prevail (see the discussion that precedes Equation (7) and also the discussion that follows Equation (35) for further details).

In this paper, we develop these ideas and build an explicit model of local hidden variables for the GHZ state of three entangled qubits. The paper is organized as follows. In Section 2, we review the argument put forward by Greenberger, Horne, and Zeilinger as a proof of the impossibility of reproducing the quantum mechanical predictions for the GHZ state within the framework of any model of local hidden variables. In Section 3, we introduce a simple, explicit model of hidden variables that overcomes this argument. In Section 4, we extend this model and discuss it in detail. Our conclusions are summarized in Section 5.

## 2. The GHZ Paradox

The Greenberger-Horne-Zeilinger spin polarization state of three entangled qubits, denoted as *A*, *B*, and *C*, is described by the quantum wavefunction:

$$|\Pi\rangle_\Phi = \frac{1}{\sqrt{2}} \left( |\uparrow\rangle^{(A)} |\uparrow\rangle^{(B)} |\uparrow\rangle^{(C)} + e^{i\Phi} |\downarrow\rangle^{(A)} |\downarrow\rangle^{(B)} |\downarrow\rangle^{(C)} \right),$$

where $\{|\uparrow\rangle, |\downarrow\rangle\}$ denotes a basis of single particle spin polarization eigenstates along its locally defined *Z*-axis. In this state, all three outcomes in every single event of a long sequence of strong spin polarization measurements performed on each one of the three qubits along their corresponding *Z*-axes must be consistently equal, either

$$S_Z^{(A)}(n) = S_Z^{(B)}(n) = S_Z^{(C)}(n) = +1,$$

or

$$S_Z^{(A)}(n) = S_Z^{(B)}(n) = S_Z^{(C)}(n) = -1,$$

for all $n \in \{1, \ldots, N\}$, with each one of the two possibilities happening with a probability of 1/2. Here *n* labels each one of the many repetitions of the experiment, and *N* is the total number of repetitions.

In fact, in the GHZ state (2) the expected average values of long sequences of strong spin polarization measurements performed along any arbitrary directions $\Omega_\alpha^{(A)}$, $\Omega_\beta^{(B)}$, $\Omega_\gamma^{(C)}$ in the XY-planes orthogonal to the local *Z*-axes are equal to zero:

$$\langle S_{\Omega_\alpha}^{(A)}(n) \rangle_{n \in \mathbf{N}} = \langle S_{\Omega_\beta}^{(B)}(n) \rangle_{n \in \mathbf{N}} = \langle S_{\Omega_\gamma}^{(C)}(n) \rangle_{n \in \mathbf{N}} = 0, \tag{2}$$

as well as their two-particles correlations:

$$\langle S_{\Omega_\alpha}^{(A)}(n) \cdot S_{\Omega_\beta}^{(B)}(n) \rangle_{n \in \mathbf{N}} = \langle S_{\Omega_\beta}^{(B)}(n) \cdot S_{\Omega_\gamma}^{(C)}(n) \rangle_{n \in \mathbf{N}}$$
$$= \langle S_{\Omega_\gamma}^{(C)}(n) \cdot S_{\Omega_\alpha}^{(A)}(n) \rangle_{n \in \mathbf{N}} = 0. \tag{3}$$

Notwithstanding, the three-particles correlation is non-zero, in general, and given by:

$$\langle S^{(A)}_{\Omega_\alpha}(n) \cdot S^{(B)}_{\Omega_\beta}(n) \cdot S^{(C)}_{\Omega_\gamma}(n)\rangle_{n\in\mathbf{N}}$$
$$= \cos\left(\Delta_{\Omega_\alpha^{(A)}} + \Delta_{\Omega_\beta^{(B)}} + \Delta_{\Omega_\gamma^{(C)}} + \Phi\right), \tag{4}$$

where $\Delta_{\Omega_\alpha^{(A)}}$, $\Delta_{\Omega_\beta^{(B)}}$ and $\Delta_{\Omega_\gamma^{(C)}}$ describe the relative orientations of each one of the measurement devices with respect to some implicit local reference directions labelled as $X$-axes, see Figure 1.

In particular, for $\Phi = 0$ the following four relationships follow:

$$\begin{array}{ccccccl}
S_X^{(A)}(n) & \cdot & S_X^{(B)}(n) & \cdot & S_X^{(C)}(n) & = +1, & n = 1, \ldots, N \\
S_X^{(A)}(m) & \cdot & S_Y^{(B)}(m) & \cdot & S_Y^{(C)}(m) & = -1, & m = 1, \ldots, M \\
S_Y^{(A)}(k) & \cdot & S_X^{(B)}(k) & \cdot & S_Y^{(C)}(k) & = -1, & k = 1, \ldots, K \\
S_Y^{(A)}(l) & \cdot & S_Y^{(B)}(l) & \cdot & S_X^{(C)}(l) & = -1, & l = 1, \ldots, L,
\end{array} \tag{5}$$

for any four sequences of strong measurements performed along directions $(X, X, X)$, $(X, Y, Y)$, $(Y, X, Y)$ and $(Y, Y, X)$.

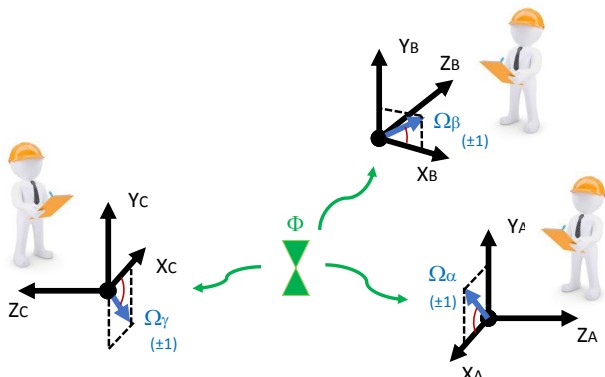

**Figure 1.** The GHZ argument implicitly requires the existence of an absolute frame of reference with respect to which it is possible to describe the polarization properties $(\pm 1, \pm 1, \pm 1)$ of the hypothetical hidden configurations of the triplets of qubits as well as the orientations $\Omega_\alpha^{(A)}$, $\Omega_\beta^{(B)}$, $\Omega_\gamma^{(C)}$ of the measurement devices that test them.

These four relationships (5) lie at the core of the Greenberger-Horne-Zeilinger paradox [8]. On the one hand, these relationships imply that we can gain certainty about the polarization properties of any of these three qubits, without in any sense, disturbing them. Thus, according to the notion introduced by Einstein, Podolsky, and Rosen [1], these polarization properties are *elements of reality* whose values must be set at the time when the three entangled particles are produced. On the other hand, this notion seems to be inconsistent: by multiplying the last three equations in (5) and assuming that all polarization components must take values either $+1$ or $-1$, we would obtain that

$$S_X^{(A)}(n) \quad \cdot \quad S_X^{(B)}(n) \quad \cdot \quad S_X^{(C)}(n) \quad = -1, \quad n = 1, \ldots, N \tag{6}$$

which is in contradiction with the first one.

This argument is widely considered the most clear-cut evidence against the possibility of giving the wavefunction (2) a statistical interpretation within the framework of a local model of hidden configurations, in which the observers are free to choose the setting of their measurements.

## 3. The Paradox Revisited

The above argument crucially relies on the implicitly assumed existence of an absolute angular frame of reference, with respect to which the polarization properties of the hidden

configurations of the triplets of entangled qubits, as well as the orientations of the measurement devices that test them, can be defined. In such an absolute frame of reference, all the polarization components of all possible hidden configurations must take a binary value, either $+1$ or $-1$, and relationships (5) immediately follow. However, as we already noticed in previous works [24–27], the existence of such an absolute angular frame of reference is not required by fundamental physical principles.

In fact, an absolute frame of reference can not be defined within the standard framework of quantum mechanics, whose predictions the models of hidden variables are aimed to reproduce. This can be readily noticed from the wavefunction that describes the GHZ state (2) in terms of the single-particle eigenstates $|\uparrow\rangle^{(A,B,C)}$, $|\downarrow\rangle^{(A,B,C)}$ of locally defined operators $\sigma_Z^{(A,B,C)}$. These eigenstates are defined only up to a phase (like any other normalized eigenvector of any linear operator) and, hence, the phase $\Phi$ in the wavefunction (2) is not, in principle, properly defined yet. In order to properly define this phase, it is necessary to set a reference setting (8) of the three measurement devices and experimentally obtain the threesome correlation between their outcomes. Only with respect to this reference setting of the three detectors, which we arbitrarily label as local $X$-axes, it is possible to properly define a subsequent rotation of any one of the devices by an angle $\Delta$, see Figure 2. Indeed, the correlation between the binary outcomes of the three measurement devices that test the GHZ state is described by a single physical degree of freedom, the angle $\Delta + \Phi$,

$$\langle S_{\Omega_\alpha}^{(A)}(n) \cdot S_{\Omega_\beta}^{(B)}(n) \cdot S_{\Omega_\gamma}^{(C)}(n) \rangle_{n \in \mathbf{N}} = \cos(\Delta + \Phi). \tag{7}$$

while the orientations of each one of the three detectors, $\Omega_\alpha^{(A)}$, $\Omega_\beta^{(B)}$ and $\Omega_\gamma^{(C)}$, cannot be independently defined in a proper sense.

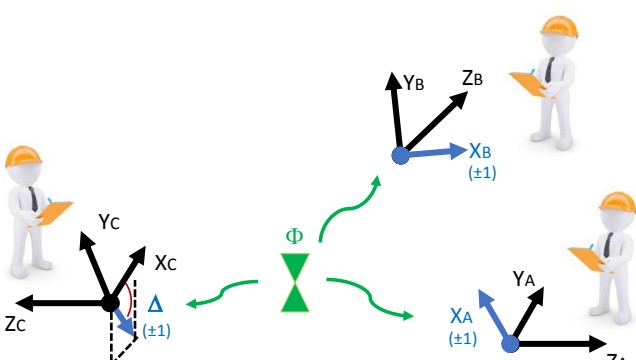

**Figure 2.** By symmetry considerations the orientation of one of the measurement devices, say *A*, can always be defined as a local *X*-axis for every one of the repetitions of the experiment. Moreover, the orientation of a second measurement device, say *B*, can also be always defined as a local *X*-axis since any rotation in it can be accounted for through the definition of the phase $\Phi$ that characterizes the source of the photons. In fact, as explained in Section 3, this is strictly necessary in order to properly define the quantum state (2). Thus, the experimental setting of the three measurement devices is described by a single angle $\Delta$, while the expected correlation between their outcomes depends only on the linear combination $\Delta + \Phi$.

In the absence of an absolute frame of reference, the polarization properties of the hidden configurations may only be properly defined with respect to the reference directions set by the orientation of the corresponding measurement devices. In particular, for the reference setting of the detectors, $\Delta = 0$, whose orientations we have arbitrarily labeled as $X$-axes, the correlation is given by:

$$\langle S_X^{(A)}(n) \cdot S_X^{(B)}(n) \cdot S_X^{(C)}(n) \rangle_{n \in \mathbf{N}} = \cos(\Phi), \tag{8}$$

and, for $\Phi = 0$ is given by

$$\langle S_X^{(A)}(n) \cdot S_X^{(B)}(n) \cdot S_X^{(C)}(n) \rangle_{n \in \mathbf{N}} = +1. \tag{9}$$

Actually, condition (8) defines the notion of *parallel* orientations of the three measurement devices. All settings for which this condition is fulfilled are physically indistinguishable through measurements performed on triplets of entangled particles in the GHZ state, and, hence, all such sets of axes are gauge equivalent.

　　Moreover, the polarization properties of the hidden configurations of the triplets of entangled particles can be properly defined only with respect to the local reference directions set by the three measurement devices. That is, the actual value $s_\Omega^{(A)}(\Omega_\alpha, \omega)$ of the polarization component of, say, particle A along some direction $\Omega$ may be, in general, a function of the reference direction $\Omega_\alpha$ set by the measurement apparatus of observer A (and, of course, also of the coordinate $\omega \in \mathcal{S}$ that labels the hidden configuration in which the system of three entangled particles occurs). This dependence does not conflict with the principle of causality, which only demands that the value of the polarization components of particle A cannot depend on the orientations of the reference directions $\Omega_\beta$, $\Omega_\gamma$ along which observers B and C choose to test their particles. Therefore, we must not restrict our models within the constraint that all polarization components of either one of the particles must take a binary value, either $+1$ or $-1$: only the polarization component of each one of the particles along the reference direction set by the orientation of the corresponding measurement device must take a binary value. That is, for all possible hidden configurations of the triplet, we must have:

$$s_{\Omega_\alpha}^{(A)}(\Omega_\alpha, \omega) = \pm 1, \quad s_{\Omega_\beta}^{(B)}(\Omega_\beta, \omega) = \pm 1, \quad s_{\Omega_\gamma}^{(C)}(\Omega_\gamma, \omega) = \pm 1, \tag{10}$$

but the polarization components along any other direction must not necessarily take either one of these two values. Indeed, the only experimental access that we can have to the spin polarization components along these other directions is through weak measurements, whose outcome can have absolute values larger and smaller than one and may even be complex [29]. In fact, weak values of physical observables are complex numbers independent of the linear dimension of the Hilbert space of the described quantum system.

　　Therefore, it is crucial to realize that in order to obtain a meaningful description of the system, we must be careful to compare magnitudes defined with respect to the same reference directions. For example, we can state that with respect to a set of *parallel* reference directions $X^{(A)}$, $X^{(B)}$ and $X^{(C)}$ defined by condition (9), the polarization components of the particles along the orthogonal directions $Y^{(A)}$, $Y^{(B)}$, $Y^{(C)}$ take values either $+i$ or $-i$, according to the relationship:

$$\begin{aligned} s_Y^{(A)}(X, \omega) &= i\; s_X^{(A)}(X, \omega), \\ s_Y^{(B)}(X, \omega) &= i\; s_X^{(B)}(X, \omega), \\ s_Y^{(C)}(X, \omega) &= i\; s_X^{(C)}(X, \omega), \end{aligned} \tag{11}$$

with $s_X^{(A)}(X, \omega) = \pm 1$, $s_X^{(B)}(X, \omega) = \pm 1$ and $s_X^{(C)}(X, \omega) = \pm 1$, see Figure 3. Therefore, the four constraints (5) become trivially identical,

$$\begin{aligned} s_X^{(A)}(X, \omega) \;\cdot\; s_X^{(B)}(X, \omega) \;\cdot\; s_X^{(C)}(X, \omega) &= +1, \\ s_X^{(A)}(X, \omega) \;\cdot\; i\, s_X^{(B)}(X, \omega) \;\cdot\; i\, s_X^{(C)}(X, \omega) &= -1, \\ i\, s_X^{(A)}(X, \omega) \;\cdot\; s_X^{(B)}(X, \omega) \;\cdot\; i\, s_X^{(C)}(X, \omega) &= -1, \\ i\, s_X^{(A)}(X, \omega) \;\cdot\; i\, s_X^{(B)}(X, \omega) \;\cdot\; s_X^{(C)}(X, \omega) &= -1. \end{aligned} \tag{12}$$

In other words, the argument put forward by Greenberger, Horne, and Zeilinger as a proof of the impossibility of reproducing the predictions of quantum mechanics for the GHZ state within the framework of a model of local hidden variables can be overcome by realizing that there does not necessarily exist an absolute frame of reference with respect to which the hidden polarization properties of the entangled particles can be defined and, in consequence, allowing their actual values to depend on the reference direction with respect to which they are described.

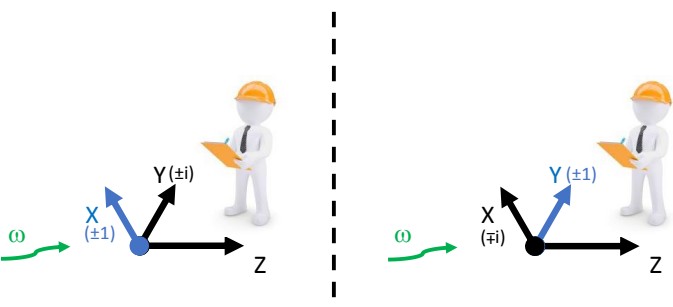

**Figure 3.** Two gauge-equivalent descriptions of the polarization properties of an incoming photon from a GHZ triplet, with respect to two different orientations of the measurement device that tests them.

## 4. A Statistical Model for the GHZ State

In this section, we build and discuss in detail an explicit statistical model of local hidden variables for the GHZ state of three entangled qubits. The model complies with the 'free-will' assumption and reproduces the quantum mechanical predictions for the average values and correlations of long sequences of strong spin polarization measurements performed on the three qubits along any three arbitrary directions.

Our statistical model consists of infinitely many possible hidden configurations continuously distributed over the unit circle $\mathcal{S}_1$, a well-defined density of probability for each one of these configurations to occur, and locally defined binary response functions that specify the outcomes that each one of these hidden configurations would produce in each one of the three measurement devices as a function of their orientations.

First, we define two sub-populations within the space of all possible hidden configurations, which we label as $\eta = \pm 1$, each one occurring with a probability of $1/2$. These two sub-populations correspond, respectively, to the two possible outcomes of the measurement performed on one of the particles, say, particle A. That is,

$$S_X^{(A)}(\eta) = s_X^{(A)}(X, \eta) = \eta. \tag{13}$$

The device that measures the polarization of particle B along an arbitrary direction orthogonal to its locally defined Z-axis fixes a reference frame of angular coordinates $\omega_B \in [-\pi, \pi)$ over the circle $\mathcal{S}_1$. We assume that the density of probability for each one of the hidden configurations to occur is given by

$$g(\omega_B) = \frac{1}{4}|\sin(\omega_B)|, \tag{14}$$

and define the outcome of the measurement on particle B as:

$$S_X^{(B)}(\omega_B, \eta) = s_X^{(B)}(X, \omega_B, \eta) = \eta \cdot S(\omega_B), \tag{15}$$

with:

$$S(x) = \text{sign}(x) = \begin{cases} +1, & \text{if } x \in (0, +\pi], \\ -1, & \text{if } x \in (-\pi, 0]. \end{cases} \tag{16}$$

Similarly, the device that measures the polarization of particle *C* along some other arbitrary direction $\Omega$ orthogonal to its locally defined *Z*-axis sets its own frame of angular coordinates $\omega_C \in [-\pi, +\pi)$ over the circle $\mathcal{S}_1$. By symmetry considerations, we demand that the outcome of this measurement be described by the same response function:

$$S_\Omega^{(C)}(\omega_C, \eta) = s_\Omega^{(C)}(\Omega, \omega_C, \eta) = \eta \cdot S(\omega_C).$$

Moreover, we impose that the two sets of angular coordinates $\omega_B$ and $\omega_C$ are related by the relationship:

$$\begin{array}{rclcl} \omega_C & = & & \omega_B', & \text{if} \quad \eta = +1, \\ \omega_C & = & \pi + & \omega_B', & \text{if} \quad \eta = -1, \end{array} \tag{17}$$

where

$$\omega_B' = L(\omega_B; \Delta + \Phi), \tag{18}$$

and

- If $\widetilde{\Delta} \in [0, \pi)$,

$$L(\omega; \widetilde{\Delta}) = \begin{cases} q(\omega) \cdot \text{arc-cos}\left(-\cos(\widetilde{\Delta}) - \cos(\omega) - 1\right), \\ \qquad\qquad \text{if} \quad -\pi \quad \leq \omega < \widetilde{\Delta} - \pi, \\ q(\omega) \cdot \text{arc-cos}\left(+\cos(\widetilde{\Delta}) + \cos(\omega) - 1\right), \\ \qquad\qquad \text{if} \quad \widetilde{\Delta} - \pi \quad \leq \omega < \quad 0, \\ q(\omega) \cdot \text{arc-cos}\left(+\cos(\widetilde{\Delta}) - \cos(\omega) + 1\right), \\ \qquad\qquad \text{if} \quad 0 \quad \leq \omega < \quad \widetilde{\Delta}, \\ q(\omega) \cdot \text{arc-cos}\left(-\cos(\widetilde{\Delta}) + \cos(\omega) + 1\right), \\ \qquad\qquad \text{if} \quad \widetilde{\Delta} \quad \leq \omega < +\pi, \end{cases} \tag{19}$$

- If $\widetilde{\Delta} \in [-\pi, 0)$,

$$L(\omega; \widetilde{\Delta}) = \begin{cases} q(\omega) \cdot \text{arc-cos}\left(-\cos(\widetilde{\Delta}) + \cos(\omega) + 1\right), \\ \qquad\qquad \text{if} \quad -\pi \quad \leq \omega < \widetilde{\Delta}, \\ q(\omega) \cdot \text{arc-cos}\left(+\cos(\widetilde{\Delta}) - \cos(\omega) + 1\right), \\ \qquad\qquad \text{if} \quad \widetilde{\Delta} \quad \leq \omega < \quad 0, \\ q(\omega) \cdot \text{arc-cos}\left(+\cos(\widetilde{\Delta}) + \cos(\omega) - 1\right), \\ \qquad\qquad \text{if} \quad 0 \quad \leq \omega < \widetilde{\Delta} + \pi, \\ q(\omega) \cdot \text{arc-cos}\left(-\cos(\widetilde{\Delta}) - \cos(\omega) - 1\right), \\ \qquad\qquad \text{if} \quad \widetilde{\Delta} + \pi \leq \omega < +\pi, \end{cases} \tag{20}$$

with

$$q(\omega) = \text{sign}((\omega - \widetilde{\Delta}) \bmod ([-\pi, \pi))),$$

and the function $y = \text{arc-cos}(x)$ is defined in its main branch, such that $y \in [0, \pi]$ while $x \in [-1, +1]$. The parameter $\widetilde{\Delta} \equiv \Delta + \Phi$ in this transformation law denotes the orientation of the measurement setting, as defined in (7) and (8).

It is straightforward to check that the density of probability (14) remains functionally invariant when described with respect to the new set of coordinates, that is,

$$g(\omega_C) = \frac{1}{4}|\sin(\omega_C)|, \tag{21}$$

since

$$|d\omega' g(\omega')| = \tfrac{1}{4}|d\omega' \sin(\omega')| = \tfrac{1}{4}|d(\cos(\omega'))| = \\ \tfrac{1}{4}|d(\cos(\omega))| = \tfrac{1}{4}|d\omega \sin(\omega)| = |d\omega g(\omega)|, \tag{22}$$

and

$$g(\pi + \omega) = \frac{1}{4}|\sin(\omega + \pi)| = \frac{1}{4}|\sin(\omega)| = g(\omega). \tag{23}$$

In fact, these equalities state in precise terms that the probability of each hidden configuration occuring does not depend on the orientation of the reference direction chosen by the observers to describe their particles, or, in other words, that our model complies with the requirements of 'free-will'.

We can now define a partition of the circle $\mathcal{S}_1$ into four disjoint regions,

$$\mathcal{S}_1 = \mathcal{I}_{++} \bigcup \mathcal{I}_{+-} \bigcup \mathcal{I}_{-+} \bigcup \mathcal{I}_{--}, \tag{24}$$

as follows:

$$\mathcal{I}_{++} = \{\omega_B : \omega_B \in (\widetilde{\Delta}, +\pi]\} = \\ = \begin{cases} \{\omega_C : \omega_C \in (0, -\widetilde{\Delta} + \pi]\}, & \text{if } \eta = +1 \\ \{\omega_C : \omega_C \in (-\pi, -\widetilde{\Delta}]\}, & \text{if } \eta = -1 \end{cases}$$

$$\mathcal{I}_{+-} = \{\omega_B : \omega_B \in (0, \widetilde{\Delta}]\} = \\ = \begin{cases} \{\omega_C : \omega_C \in (-\widetilde{\Delta}, 0]\}, & \text{if } \eta = +1 \\ \{\omega_C : \omega_C \in (-\widetilde{\Delta} + \pi, \pi]\}, & \text{if } \eta = -1 \end{cases}$$

$$\mathcal{I}_{--} = \{\omega_B : \omega_B \in (\widetilde{\Delta} - \pi, 0]\} = \\ = \begin{cases} \{\omega_C : \omega_C \in (-\pi, -\widetilde{\Delta}]\}, & \text{if } \eta = +1 \\ \{\omega_C : \omega_C \in (0, -\widetilde{\Delta} + \pi]\}, & \text{if } \eta = -1 \end{cases}$$

$$\mathcal{I}_{-+} = \{\omega_B : \omega_B \in (-\pi, \widetilde{\Delta} - \pi]\} = \\ \begin{cases} \{\omega_C : \omega_C \in (-\widetilde{\Delta} + \pi, \pi]\}, & \text{if } \eta = +1 \\ \{\omega_C : \omega_C \in (-\widetilde{\Delta}, 0]\}, & \text{if } \eta = -1 \end{cases}$$

where we have assumed without any loss of generality that $0 \leq \widetilde{\Delta} \leq \pi$.

In each one of these four segments, the two measurements are fully correlated or anti-correlated:

- If $\eta = +1$,

$$\begin{aligned} S_X^{(B)}(\omega_B, \eta) \cdot S_\Omega^{(C)}(\omega_C, \eta)\Big|_{\mathcal{I}_{++} \cup \mathcal{I}_{--}} &= +1, \\ S_X^{(B)}(\omega_B, \eta) \cdot S_\Omega^{(C)}(\omega_C, \eta)\Big|_{\mathcal{I}_{+-} \cup \mathcal{I}_{-+}} &= -1, \end{aligned} \tag{25}$$

- If $\eta = -1$,

$$\begin{aligned} S_X^{(B)}(\omega_B, \eta) \cdot S_\Omega^{(C)}(\omega_C, \eta)\Big|_{\mathcal{I}_{++} \cup \mathcal{I}_{--}} &= -1, \\ S_X^{(B)}(\omega_B, \eta) \cdot S_\Omega^{(C)}(\omega_C, \eta)\Big|_{\mathcal{I}_{+-} \cup \mathcal{I}_{-+}} &= +1, \end{aligned} \tag{26}$$

It is straighforward to notice that

$$\mu\left(\mathcal{I}_{++} \bigcup \mathcal{I}_{--}\right) - \mu\left(\mathcal{I}_{+-} \bigcup \mathcal{I}_{-+}\right) = \cos(\widetilde{\Delta}), \tag{27}$$

where $\mu(\cdot)$ denotes the normalized measure over the circle according to the probability density distribution (14). Hence,

- Over the sub-population of states with $\eta = +1$,

$$\langle S_X^{(B)} \cdot S_\Omega^{(C)} \rangle = \cos(\widetilde{\Delta}), \tag{28}$$

- Over the sub-population of states with $\eta = -1$,

$$\langle S_X^{(B)} \cdot S_\Omega^{(C)} \rangle = -\cos(\widetilde{\Delta}). \tag{29}$$

Therefore, over the whole population the two measurements are totally uncorrelated,

$$\langle S_X^{(B)} \cdot S_\Omega^{(C)} \rangle = 0, \tag{30}$$

since each one of the two sub-populations $\eta = +1$ and $\eta = -1$ happens with probability $1/2$. The same is true for the same reason for the correlation between the outcome of the measurement on particle $A$ and any of the other two:

$$\langle S_X^{(A)} \cdot S_X^{(B)} \rangle = \langle S_X^{(A)} \cdot S_\Omega^{(C)} \rangle = 0. \tag{31}$$

Furthermore, the three-particles correlation is given by:

$$\langle S_X^{(A)} \cdot S_X^{(B)} \cdot S_\Omega^{(C)} \rangle = \cos(\widetilde{\Delta}), \tag{32}$$

which reproduces the quantum mechanical prediction (7) for the GHZ state.

Let us remark that in the model that we have described here, similar to quantum formalism, the orientation of two of the three measurement devices sets a reference frame with respect to which the orientation of the third device is described; see Figure 2. Therefore, it does not make sense to compare two different orientations for the reference setting since they are physically indistinguishable and, hence, the orientation of the reference setting is a spurious gauge of degree of freedom. This is the ultimate reason that allows the set of angular coordinates over the circle $\mathcal{S}_1$ to acquire (due to a holonomy) a non-zero geometric phase $\alpha \neq 0, \pi$ through certain cyclic transformations (19) and (20):

$$\mathcal{L}_{-\Delta} \circ \mathcal{L}_{\Delta+\Phi} \circ \mathcal{L}_{-\Phi} = \mathcal{L}_\alpha \neq \mathbb{I}, -\mathbb{I}. \tag{33}$$

The possible appearance of a geometric phase in closed loops of gauge transformations is well-known, also in classical physics. A particularly beautiful example is the gauge theory of swimming at low Reynolds numbers described in ref. [30].

Before closing this section, let us stress that Equations (17) and (18) are coordinates transformations and do not introduce any non-local interaction between the detectors. In order to clarify this issue consider a source that produces pairs of macroscopic arrows parallel to each other and randomly oriented within a locally defined XY plane. The twin arrows are then parallel-transported in opposite directions along the Z axis towards two distant detectors, each one of them consisting of an arrow that can also be arbitrarily oriented within their local XY plane. Upon arriving at their respective detectors, the relative orientation of each one of the incoming arrows is described with respect to the orientation of the corresponding detector, and a local response is produced according to (16). Obviously, for every pair of incoming twin arrows, the following relationship must hold:

$$\omega' = \omega - \widetilde{\Delta}, \tag{34}$$

where $\omega$ and $\omega'$ are the relative angles between the orientations of the incoming arrows and their corresponding detectors, and $\widetilde{\Delta}$ is the relative angle between the two detectors. This relationship (34) does not introduce any non-local interaction between the detectors,

since it is dictated by the euclidean structure of the macroscopic space, and, therefore, it is fulfilled no matter who decides how to orient the detectors or when these decisions are taken. Furthermore, the response of each one of the detectors given by (16) depends only on the orientation of the incoming arrow with respect to the local detector and does not depend either on the relative orientation between the two detectors or on the orientation of the other arrow with respect to the other detector.

Equations (17) and (18) are nothing but a non-linear generalization of the Euclidean relationship (34), and it simply means that the entangled particles might carry with them a non-euclidean metric. In this sense, it is useful to think about Equations (17) and (18) as somehow similar to the Lorentz transformation that relates, for example, the frequencies $\nu$ and $\nu'$ of a signal emitted by a source towards two detectors moving with relative velocity $V$,

$$\nu' = L(\nu; V). \tag{35}$$

Obviously, this non-linear relationship does not violate Einstein's principle of causality since it is dictated by the Minkowski metric of space-time, from which the very notion of causality stems.

As a last final comment, let us remind ourselves again that Bell's definition of locality arose as a result of the intent by Bell to formulate Einstein's principle of causality in a way fit to prove his renowned Bell's theorem. Therefore, wherever the notion of Bell's locality disagrees with Einstein's principle of causality, the latter must prevail.

## 5. Discussion

We have shown in this paper that the argument behind the renowned GHZ paradox crucially relies on an implicit assumption that is not required by fundamental physical principles and, therefore, can be overcome by giving up this unnecessary requirement. Namely, the argument put forward by Greenbereger, Horne, and Zeilinger thirty years ago [8] implicitly assumes that there exists an absolute angular frame of reference with respect to which we can define the polarization properties of the hypothetical hidden configurations of the entangled qubits, as well as the orientations of the measurement devices that test them. However, we have remarked in this and previous papers [24–27] that in order to properly define the phase $\Phi$ that characterizes the state (2) of the triplets of entangled qubits, it is necessary to fix an arbitrary reference setting of the measurement devices that test their polarizations. Only with respect to this reference setting can one properly define a subsequent relative rotation $\Delta$ of one of the devices, see Figure 2, while the orientation of the reference setting is a spurious gauge degree of freedom. In the absence of an absolute frame of reference, the polarization properties of each of the qubits can be properly defined only with respect to the orientation of the measurement device that tests them.

With these observations in mind, we have built an explicit statistical model of local hidden variables for the GHZ state of three entangled qubits that reproduces the predictions of quantum mechanics and complies with the 'free-will' assumption. The model, thus, completes—in the EPR sense—the quantum description of the GHZ state. This model closely resembles the model of hidden variables for the Bell polarization states of two entangled qubits that we recently described in [24–27].

Since these models were designed to reproduce the predictions of quantum mechanics for the Bell and GHZ experiments, they cannot be experimentally favored or disfavored against the quantum formalism through their predictions for these experiments. Further work is needed in order to develop this statistical framework and maybe find ways to test it against quantum mechanics, but this is beyond the scope of the present papers. The aim of these models at this stage is to explore the possibility that the strongly well-established quantum formalism could not be the ultimate framework for describing the fundamental building blocks of Nature and their interactions, overcoming a belief widely held by the physics community for over half a century. An underlying statistical framework

would provide a physically intuitive interpretation of the quantum formalism and a better understanding of quantum phenomena.

Finally, it is worth stressing once more that the existence of an absolute frame of reference is neither demanded nor guaranteed by fundamental physical principles or any experimental evidence, and, therefore, it is at best a working assumption. However, according to the conclusions reached in this paper and [24–27], this working assumption lies at the core of the impossibility noticed by the Bell theorem, the GHZ theorem, and other renowned theorems to accommodate together within the quantum formalism some of the most fundamental physical notions. On the other hand, these difficulties can be easily overcome by lifting this working assumption. Therefore, the latter might even be considered a favored option since the apparent emergence of an absolute frame in the macroscopic world can also be easily understood [24].

**Funding:** This research received no external funding.

**Data Availability Statement:** Not applicable.

**Conflicts of Interest:** The author declares no conflict of interest.

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
