# Peer review of "The GHZ Theorem Revisited within the Framework of Gauge Theory"

_symmetry, doi:10.3390/sym15071327_

Round 1
Reviewer 1 Report
This is a very nice and important paper devoted to the foundations of quantum mechanics. The author presents a method (which was also discussed in his previous papers) to overcome the conclusions about nonexistence of hidden variables which seem to follow from Greenberger-Horne-Zeilinger (GHZ) version of of the Einstein-Podolsky-Rosen (EPR) paradox . This method is founded on the observation that an implicit assumption of GHZ experiment that there exists an absolute angular frame of reference with respect to which one can define the polarization properties of the hidden configurations and the orientations of the measurement is not necessary true. Of course, the question is whether the author is right but the conclusive answer can be done only by further experimental and theoretical investigations. Independently of this the paper is very interesting and I warmly recommend it for publication.
Author Response
Dear Editor,
I wish to thank the referee for his comments. He is correct that further theoretical and experimental work is needed in order to decide if the statistical framework developed in this paper indeed describes how Nature actually works, but this is beyond the scope of the present paper. The aim of this paper is only to show that such a framework is not necessarily ruled out and it is indeed possible.
Reviewer 2 Report
The authors present their study on the GHZ theorm, which give up the assumption of the absolute angular frame of reference and replaced by a gauge degree of freedom. It is an interesting and meaningful result for understauding quantum nolocality. The manuscript may be considered publication in Symmetry.
Author Response
Dear Editor,
I wish to thank the referee for his comments.
Reviewer 3 Report
The paper is interesting and might be published after major revision when
the following issues are clarified.
1. What is wrong with Bell's notion of locality? Please, provide examples where this notion is in disagreement with the Einstein's principle of causality.
2.Statistical model for the GHZ state: what are the specific experimental facts for its favour?
3. Experimental status of the statistical interpretation of QM?
4. What are the experimental grounds for the assumption on the absence of the absolute angular reference frame?
5. n and N: not explained quantities, line 119?
Author Response
Dear Editor,
I wish to thank the referee for his comments and questions. Please find enclosed a point-by-point response to each one of them.

Round 2
Reviewer 3 Report
I have found the answers detailed and i think that the revised
version may be published.